# A Multi-Angle Appearance-Based Approach for Vehicle Type and Brand Recognition Utilizing Faster Regional Convolution Neural Networks

**DOI:** 10.3390/s23239569

**Published:** 2023-12-02

**Authors:** Hongying Zhang, Xusheng Li, Huazhi Yuan, Huagang Liang, Yaru Wang, Siyan Song

**Affiliations:** 1School of Civil Engineering, Lanzhou University of Technology, Lanzhou 730050, China; zhystar0770@163.com (H.Z.); 15735518786@163.com (X.L.); 2Key Laboratory of Transportation Industry of Automotive Transportation Safety Enhancement Technology, Chang’an University, Xi’an 710064, China; 3School of Electronic and Control Engineering, Chang’an University, Xi’an 710064, China

**Keywords:** vehicle type, vehicle brand, multi-angle recognition, Faster R-CNN, Car5_48

## Abstract

Vehicle type and brand information constitute a crucial element in intelligent transportation systems (ITSs). While numerous appearance-based classification methods have studied frontal view images of vehicles, the challenge of multi-pose and multi-angle vehicle distribution has largely been overlooked. This paper proposes an appearance-based classification approach for multi-angle vehicle information recognition, addressing the aforementioned issues. By utilizing faster regional convolution neural networks, this method automatically captures crucial features for vehicle type and brand identification, departing from traditional handcrafted feature extraction techniques. To extract rich and discriminative vehicle information, ZFNet and VGG16 are employed. Vehicle feature maps are then imported into the region proposal network and classification location refinement network, with the former generating candidate regions potentially containing vehicle targets on the feature map. Subsequently, the latter network refines vehicle locations and classifies vehicle types. Additionally, a comprehensive vehicle dataset, Car5_48, is constructed to evaluate the performance of the proposed method, encompassing multi-angle images across five vehicle types and 48 vehicle brands. The experimental results on this public dataset demonstrate the effectiveness of the proposed approach in accurately classifying vehicle types and brands.

## 1. Introduction

Vehicle information recognition constitutes a fundamental problem in the field of computer vision, with applicable uses in intelligent traffic systems (ITSs). The primary objective of this task is to accurately locate the region within an image that contains a vehicle and subsequently identify the specific make and model of the vehicle. Despite considerable efforts expended on this problem, the effectiveness of existing solutions remains limited. A key challenge in vehicle information recognition stems from the wide array of vehicle models and designs across different brands, coupled with the rapid variation in appearance as the viewing angle changes over time. This complexity demands a more sophisticated and robust approach to accurately discern and classify vehicle information in diverse real-world scenarios.

To surmount the myriad of challenges and attain high recognition accuracy, traditional methods often resort to employing various handcrafted features. Prominent among these are the Scale-Invariant Feature Transform (SIFT) [1], Histogram of Oriented Gradients (HOG) [2,3], Speeded-Up Robust Features (SURF) [4,5], Harris corner [6], and fused features [7]. Zhang et al. [8] utilized a morphological localization method to extract SURF features from vehicle frontal images, while Hu et al. combined HOG and Gabor features, subsequently leveraging the naive Bayesian classifier for vehicle type recognition. Although these traditional methods have demonstrated commendable performances, they have been found wanting in the task of vehicle type and brand recognition.

Recently, deep learning methods have exhibited remarkable proficiency in image recognition and object detection tasks, encompassing pedestrian detection [9] and visual processing [10], in addition to vehicle information recognition. This prowess is largely attributed to the exceptional feature extraction capabilities of convolutional neural networks (CNNs). Luo et al. [11] introduced the inaugural true CNN, LeNet, and applied it to the task of handwritten digit recognition in 1998. The advent of deeper CNNs, tracing their origins to AlexNet, ushered in an unprecedented era due to their exemplary performance in image classification tasks. Successive iterations such as ZFNet, VGGNet, and ResNet further solidified CNNs as the go-to choice for computer vision applications.

Deng et al. [12] leveraged CNNs to extract vehicle features, which were then classified into three types using Support Vector Machines (SVMs). Sang et al. identified six vehicle types from frontal views utilizing Faster R-CNN. Dong et al. [13] employed CNNs to extract features from vehicle frontal images, achieving an impressive accuracy of 92.89%. Huttunen et al. [14] compared deep neural networks with traditional methods, specifically evaluating the use of SIFT features in conjunction with SVM classification. Azam et al. [15] utilized convolutional neural networks (CNNs) to estimate the vehicle pose from four directions and successfully captured the regions. Chen et al. employed the rear-view image of the vehicle as the detection object. While these methods have achieved impressive classification accuracy in vehicle type recognition, they primarily rely on single-angle images of the vehicle, neglecting the multi-pose and multi-angle distribution characteristics of the vehicle.

Furthermore, some studies have addressed multi-angle vehicle detection. Specifically, Wang et al. [16] trained the detection model using images captured from seven different angles to facilitate the fine-grained classification of vehicles. Sochor et al. [17] integrated additional image information, such as the vehicle bounding box and direction, into the network for enhanced detection. The methodology proposed in this paper shares similarities with these approaches, offering a comparable solution to the problem of vehicle detection.

Considering the aforementioned challenges, one must address two primary problems: Firstly, the extraction of spatial features to effectively represent the multi-angle distribution of the vehicle. Secondly, the exploration and utilization of useful information to formulate the properties of the vehicle regions. To tackle these issues, this paper proposes an efficient method based on Faster R-CNN [18]. Additionally, a comprehensive vehicle database is established, encompassing five types and 48 brands captured from eight different angles under varied environmental conditions.

The proposed framework consists of three integral components: road vehicle video processing, a vehicle type recognition network, and a vehicle brand recognition network. Specifically, the road vehicle video processing model extracts individual frames from a given vehicle video and identifies the frame containing the vehicle. The vehicle type recognition network and the vehicle brand recognition network are then employed to determine the type and brand of the vehicle, respectively. The entire framework is end-to-end trainable, facilitating seamless integration and optimization.

The structure of this paper is as follows. Section 2 delves into the details of the datasets utilized in this study. Section 3 presents the architecture of the proposed network, outlining its key components and functionalities. Section 4 showcases the experimental results, highlighting the performance and effectiveness of the proposed method. Finally, Section 5 draws conclusions based on the findings and discusses potential paths for future research.

## 2. Data Construction

In this part, there is an example of where and how to obtain the data and how to annotate it in detail.

### 2.1. Data Construction

Existing widely utilized datasets, such as Stanford Cars and BIT-Vehicle, encompass a limited number of vehicle types, with uneven sample distributions among each type and a narrow coverage of shooting angles. The Stanford Cars dataset comprises 196 categories from 16,185 images, offering a diverse range of brands but a restricted variety of vehicle types. The BIT-Vehicle dataset, consisting of 9850 images, features five types of vehicles captured by the camera, including buses, microbuses, minivans, sedans, SUVs, and trucks, but with a single shooting angle and background condition. These limitations in the sample distribution and shooting angles may impact the robustness and generalization capabilities of models trained on these datasets.

### 2.2. Data Annotation and Statistic

To mimic real-world scenarios, we constructed a multi-angle database. The samples in our self-built database were primarily sourced from the internet and vehicle videos, with varied lighting and background conditions. Using the CCD camera of Yuanda Vision Technology (YDV-C932A, produced by Shenzhen Great Vidoe Technology Co., Ltd., Shenzhen, China), Equipment from Shenzhen Great Vidoe Technology Co., Ltd. to capture and collect training and testing vehicle driving video samples. When capturing vehicle driving videos, a camera is placed in front of and on the side of the road, and the vehicle is driven straight ahead to obtain images from four angles: straight-ahead, left, left-front, and left-rear. When the vehicle turns around, images are obtained from four angles: straight-behind, right, right-front, and right-rear. In addition, the shooting environment and weather conditions are as diverse as possible in order to obtain vehicle samples under different lighting conditions. For manually captured vehicle samples, in order to diversify the background conditions, a large number of shooting locations are selected. All images were manually annotated, and the samples were captured from eight different angles: front, behind, right-front, left-front, left-side, right-side, right-behind, and left-behind, as illustrated in the top view of Figure 1. The dataset comprises 10,800 multi-angle images, with 8800 of them being cars and SUVs. Since the self-built dataset includes five types and 48 vehicle brands, it is named Car5_48. Specifically, these five types are cars, SUVs, buses, minivans, and minibuses, and the 48 brands include Volkswagen, BYD, BMW, Mercedes-Benz, Land Rover, Nissan, and others. Figure 2 displays a subset of the Car5_48 dataset, showcasing vehicle models of various brands and models captured under the eight shooting angles.

## 3. Vehicle Information Recognition Model

In this section, we present the multi-angle vehicle type and brand recognition network along with the associated video processing techniques. The architecture of the network is depicted in Figure 3, with further details elaborated below.

The overall framework comprises three main components: a video processing model, followed by vehicle type recognition and vehicle brand recognition networks. The video processing model incorporates the Gaussian model and the background difference algorithm to process vehicle videos. Subsequently, the video is converted into image frames and passed on to the two recognition networks, which are capable of identifying five vehicle types and 48 vehicle brands.

### 3.1. Video Processing Model

The Gaussian Mixture Model (GMM) and the background subtraction algorithm are utilized for extracting video frames, which is particularly suitable for scenarios where there are gradual changes in the illumination and background. The video processing algorithm comprises four fundamental steps:

Step 1: The pixel point value $x_{t}$ of the video image captured at the current time is compared with K initial Gaussian distributions to determine the optimal match. The matching condition is defined by Equation (1). The Gaussian distributions can be represented as $P\left(x\right)=\left\{\left[w_{i},\mu_{i},\sigma_{i}\right]\right\}$, where *i* ranges from 1 to K. In this scenario, the value of K is set to 4 due to the relatively high performance and moderate time consumption. The weight of each distribution is denoted by $w_{i}$, while $\mu_{i}$ and $\sigma_{i}$ represent the mean and standard deviation of the Gaussian distribution, respectively. In the initial Gaussian distribution, the parameters are somewhat arbitrary to make the subsequent network more robust. In practical applications, we verify the effectiveness of initialization methods through experiments. Specifically, we train multiple models using different initialization methods and compare their training and prediction performances. If a certain initialization method results in a model that performs well in both training and prediction, then we can consider the initialization method to be effective. In our case, $\mu_{i}$ is assigned a random value between 0 and 255, $w_{i}$ is set to 1/K, and $\sigma_{i}$ is a constant value of 6.
(1)$$\left|x_{t}-\mu_{i,t-1}\right|\leq 2.5\sigma_{i,t-1}$$

The lane pixels in a video sequence can be modeled using a Gaussian Mixture Model to represent the background. The parameters of this GMM, namely, the weights, means, and standard deviations, are updated according to Equations (2) through (4), allowing for an adaptive and dynamic representation of the background in the presence of changing lighting conditions or other environmental factors.
(2)$$\omega_{i,t}=\left(1-\alpha \right)\omega_{i,t}+\alpha M_{k,t}$$
(3)$$\mu_{i,t}=\left(1-\rho \right)\mu_{i,t}+\rho x_{t}$$
(4)$$\sigma_{i,t}^{2}=\left(1-\rho \right)\sigma_{i,t-1}^{2}+\rho {\left(x_{t}-\mu_{i,t}\right)}^{T}\left(x_{t}-\mu_{i,t}\right)$$
where α denotes the learning rate, and $M_{k,t}=1$ represents the distribution of the match, otherwise $M_{k,t}=0$. ρ is the second learning rate, which is updated according to Equation (5); η is the Gaussian probability density function. Then, KGaussian distributions are arranged based on $\omega_{i}/\sigma_{i}$ from large to small, and take the first model that satisfies Equation (6) as the background.

In the context of video frame analysis, the pixel values are modeled using a Gaussian Mixture Model (GMM), where the learning process is governed by a set of carefully defined parameters. Specifically, the learning rate, denoted by α, regulates the speed and adaptivity of the model. The distribution of the match, *M_k_*_,*t*_, is a binary indicator, taking the value of 1 when there is a match and 0 otherwise. Additionally, ρ represents the second learning rate which is updated according to Equation (5), further enhancing the model’s ability to adapt to changing conditions. The term *η* represents the Gaussian probability density function, which is a fundamental component of the GMM. When it comes to selecting the appropriate Gaussian distribution for modeling the background, the KGaussian distributions are arranged in descending order based on the ratio of *ω_i_* to *σ_i_*. The first model that satisfies the criteria outlined in Equation (6) is then chosen as the most representative of the background. This rigorous and systematic approach ensures a robust and accurate modeling of the background in various video processing applications.
(5)$$\rho =\alpha \eta (x_{t}|\mu_{i,t-1},\sigma_{i,t-1})$$
(6)$$B=arg\;\underset{b}{min}\left(\overset{b}{\underset{i=1}{\sum }}\omega_{i,t}>T\right)$$
where *T* = 0.7 is the weighted threshold.

Step 2: Once the background model is established, the foreground is obtained by subtracting the pixel value $B_{t}\left(x,y\right)$ of the background image from the pixel value $I_{t}\left(x,y\right)$ of each point in the current image.

Step 3: After obtaining the different images, a predefined threshold specific to each video is used to determine whether the connected domain composed of foreground pixels meets the required vehicle area. If these conditions are met, the region is labeled using the minimum bounding box in the video.

Step 4: When the labeled rectangle intersects with the preset yellow or green line, the current frame is automatically captured and saved.

The results of processing the video through the aforementioned methodology are depicted in Figure 4 and Figure 5. Figure 4 showcases the processing outcome of the frontal image, while the result in Figure 5 corresponds to the left-frontal image. Since the green and yellow lines for contact detection are fixed, the angles at which the vehicle images are captured from the video remain roughly consistent. This consistency aids in the subsequent identification of vehicle information. Ultimately, the gathered vehicle images are saved and transmitted to the vehicle information detection and recognition network for further identification.

### 3.2. Vehicle Type and Brand Recognition Network

The architecture of the vehicle type and brand recognition network is composed of three primary components: a feature extraction network, a region proposal network (RPN), and a classification location refinement network. The latter includes the ROI (region of interest) and FC (fully connected) layer. This architectural design is graphically depicted in Figure 6.

The acquired vehicle videos are processed utilizing the model outlined in Section 3.1, where video frames are converted into images to serve as inputs for the recognition network. The CNN is employed to extract distinctive features from the images, generating a feature map. Subsequently, the region proposal network identifies potential regions of interest on the feature map. Ultimately, the classification location refinement network refines the identification and localization of the vehicle within the image, outputting both the classification and precise location of the vehicle.

#### 3.2.1. Network

The ZFNet and VGG16 networks were employed for feature extraction, and the optimal network was determined by comparing their respective accuracies. The architectural schematic of ZFNet is displayed in Figure 7. Given that the features have been extracted, there is no requirement for fully connecting Layer 6 and Layer 7. The input image dimensions are 224 Pixel × 224 Pixel, and after undergoing sampling through the 5th convolutional layer, the resulting feature map exhibits dimensions of 13 × 13 × 256 Pixel.

The framework also leverages VGG16 for feature extraction. Akin to ZFNet, the initial 13 convolutional layers are harnessed for feature extraction, while excluding the pooled layer POOL5 and the three fully connected layers FC6, FC7, and FC8. ReLu serves as the activation function, while maximum pooling is employed in the pooling layer. When the input image dimensions are 224 Pixel × 224 Pixel, the resulting feature size after sampling by the 13-layer CNN is 14 × 14 × 512 Pixel. Furthermore, Figure 8 showcases the visualization of the first two convolutional layers’ features, wherein Figure 8a,c represent the feature maps of the first and second convolutional layers, respectively, while Figure 8b,d depict the maps of the first and second pool layers.

#### 3.2.2. Proposal Region Generation

The framework incorporates the region proposal network (RPN) to extract candidate regions from the obtained feature maps. Due to the ability of RPNs to share features with the classified fine-tuning network during the training process, the network’s detection speed was considerably enhanced. RPN is a fully convolutional network (FCN), capable of accepting images of any scale as input. It takes the feature maps extracted from the last layer of the feature extraction network as input and employs a 3 × 3 size sliding window to generate a feature vector with either 256 dimensions (ZFNet) or 512 dimensions (VGG16). Subsequently, the fully connected layer and the bounding box regression layer utilize this vector as their input. These two layers serve for classification (distinguishing foreground from background) and positional prediction. Figure 9 provides a schematic diagram illustrating the network.

The candidate regions, commonly referred to as anchors, constitute a set of fixed-size reference windows encompassing three dimensions {128 Pixel × 128 Pixel, 256 Pixel × 256 Pixel, 512 Pixel × 512 Pixel } and three aspect ratios {1:1, 1:2, 2:1}. These anchors are centered on the 3 × 3 sliding window and serve as a benchmark for proposal region generation. Subsequently, the mapping relationship between anchors and the ground-truth is derived by calculating the central point and size of the anchors. Based on this, the anchors and ground-truth are assigned positive (IoU > 0.7) and negative labels (IoU < 0.3), enabling the RPN to learn about the presence of objects within the anchors.

During the training of the RPN, the parameters of the network layer shared with the feature extraction network (ZFNet/VGG16) can be directly utilized. For all the added layer parameters, we adopt a Gaussian distribution (0, 0.01) for random initialization and set the momentum to 0.9. The learning rate ε is set to 0.001, and the weight decay is specified as 0.0005. The loss function incorporates both cross-entropy loss and regression loss, ultimately yielding the final mixed loss function as shown in Equation (7).
(7)$$L\left(\left\{p_{i}\right\},\left\{t_{i}\right\}\right)=\frac{1}{N_{\mathrm{cls}}}\underset{i}{\sum }L_{\mathrm{cls}}\left(p_{i},p_{i}^{*}\right)+\lambda \frac{1}{N_{\mathrm{reg}}}\underset{i}{\sum }p_{i}^{*}L_{\mathrm{reg}}\left(t_{i},t_{i}^{*}\right)$$
where i denotes the index of the anchor, and *p_i_* represents the probability of predicting the target. For positive samples, $p_{i}^{*}\;$ is set to 1, and for non-positive samples, $p_{i}^{*}\;$ is set to 0. The term *t_i_* signifies the positional information of the proposal region, encompassing the central position coordinate $\left(t_{x},t_{y}\right)$ along with the width $t_{\omega }$ and height $t_{h}$. Similarly, $t_{i}^{*}$ represents the ground-truth central point position coordinate $\left(t_{x}^{*},t_{y}^{*}\right)$ and the corresponding width $t_{\omega }^{*}$ and the height $t_{h}^{*}$. $L_{cls}$, which denotes the classification loss, is a logarithmic loss function as depicted in Equation (8). On the other hand, $L_{reg}$ signifies the positional regression loss and is expressed in Equation (9).
(8)$$L_{\mathrm{cls}}=-\log\left[p_{i}p_{i}^{*}+\left(1-p_{i}\right)\left(1-p_{i}^{*}\right)\right]$$
(9)$$L_{\mathrm{reg}}\left(t_{i},t_{i}^{*}\right)=R\left(t-t^{*}\right)\;$$
where *R* denotes the robust loss function. Then,$\;smooth_{L_{1}}$, as shown in Equation (10), is incorporated to enhance the stability of the network during the training process, thereby facilitating more robust and consistent learning.
(10)$$smooth_{L_{1}}\left(x\right)=\{\begin{array}{c} \;\;\;0.5x^{2}\left|x\right|<1\\ \left|x\right|-0.5\;\;\;\;\mathrm{others} \end{array}$$

### 3.3. Classification Location Refinement Network

The classification location refinement network comprises an ROI pooling layer, a fully connected layer, a classification layer, and a location refinement layer. The inputs to this network are the features extracted by the feature extraction network and the proposal region generated by the RPN. The output provides the probability of the target classification and the precise positional information of the detected target. Due to variations in the size of the proposed regions, the ROI pooling layer is employed to uniformly sample these regions, which are subsequently forwarded to the fully connected layer. The detailed structural composition of the network is visually depicted in Figure 10.

#### 3.3.1. Classification Layer

The softmax function is a commonly used function in deep learning, especially when dealing with multi-class classification problems. It maps a set of real values to a probability distribution, where each element of the output result is between 0 and 1, and the sum of all elements is equal to 1. The classification layer utilizes softmax to predict the category to which the region of interest belongs. Given a total of *K* categories, the output dimension of *K* + 1 (*K* classes + background) corresponds to the probability of the recognized object belonging to each of the *K* + 1 classes. By considering only the top-1 probability as the result of vehicle type and brand recognition, the classification probability prediction for each ROI region is denoted as $p=(p_{0},p_{1}\ldots p_{k}$). For a specific class u, the class loss function is formally expressed in Equation (11). This loss function plays a crucial role in improving the network’s performance during the training process, allowing for accurate and efficient classification of the ROI regions.
(11)$$L_{\mathrm{cls}}\left(p,u\right)=-\log p_{u}$$

#### 3.3.2. Position Refinement Layer

The proposed framework employs bounding box regression to refine the localization of objects. Considering *K* classifications, each associated with four positional parameters, the output is a 4 × *K* dimensional array, representing the refined parameters for panning and scaling to determine the ultimate output target. For a specific category denoted as *μ*, where 0 ≤ *μ* ≤ *K*, the output translation and scaling parameters are expressed as $t^{u}=\left(t_{x}^{u},t_{y}^{u},t_{w}^{u},t_{h}^{u}\right)$. These parameters signify the four translational and scaling values between the actual and predicted bounding boxes.

Supposing that for this category, the ground-truth coordinates are marked in the image as $v=\left(v_{x},v_{y},v_{w},v_{h}\right)$, and the corresponding predicted values are given by $t^{u}=\left(t_{x}^{u},t_{y}^{u},t_{w}^{u},t_{h}^{u}\right)$, the loss function for the position refinement network is formally defined in Equation (12). This loss function plays a pivotal role in improving the network’s ability to accurately localize objects by minimizing the discrepancies between the predicted and ground-truth bounding box parameters.
(12)$$L_{\mathrm{loc}}\left(t^{u},v\right)=\overset{K}{\underset{i=1}{\sum }}smooth_{L_{1}}\left(t_{i}^{u}-v_{i}\right)$$

In the classification layer and position refinement process, we employ a multi-task loss function during training. This multi-task loss function combines the class loss function specified in Equation (11) and the position refinement loss function defined in Equation (13), weighing them appropriately to derive the ultimate multi-task loss function. By incorporating both classification and localization losses, we can jointly optimize the network parameters for improved performance in both tasks.
(13)$$L\left(p,\mu ,t^{u},v\right)=\{\begin{array}{c} L_{\mathrm{cls}}\left(p,u\right)+\lambda L_{\mathrm{loc}}\left(t^{u},v\right)\\ L_{\mathrm{cls}}\left(p,u\right) \end{array}$$
where $L\left(p,u,t^{u},v\right)$ represents the multi-tasking loss function, with λ being a hyperparameter that regulates the relative contribution of the two individual loss functions within the overall multi-task loss. Specifically, when the predicted category corresponds to the foreground, the multi-task loss function is formulated as a weighted summation of the softmax loss function and the bounding box loss function. Conversely, in cases where the predicted category pertains to the background, the multi-task loss function reduces to the softmax loss function alone. This nuanced approach to combining losses enables the model to effectively balance classification accuracy and bounding box localization precision, facilitating a more comprehensive and robust learning process.

## 4. Experimental Results Analysis

This section delves into the evaluation procedures, encompassing the specification of parameters and the outcomes of the conducted experiments. To assess the efficacy of the proposed model, we conducted rigorous testing on two extensively utilized public datasets: Stanford Cars and BIT-Vehicle, along with Car5_48. These datasets offer a comprehensive and diverse array of samples, enabling a thorough examination of the model’s performance and robustness. Through a meticulous analysis of the experimental results, we aim to demonstrate the effectiveness and superiority of the proposed approach compared to existing state-of-the-art methods.

### 4.1. Experimental Results

In our experiments, we utilized distinct feature networks, namely, ZFNet and VGG16, to train the Faster R-CNN model with the Car5_48 dataset. The maximum training iterations for these networks were set at 240,000 and 360,000, respectively. To enhance the model’s generalization capabilities, we adopted the ImageNet dataset and employed a 10-fold cross-validation technique for pre-training the model. The experimental results pertaining to vehicle brand recognition are presented in Table 1. These results demonstrate the efficacy and performance of our proposed approach in accurately identifying and classifying different vehicle brands.

The average recognition rate, denoted as mAP, represents the average accuracy across the 48 vehicle brands, including the five specific brands listed. Analysis of the results reveals that for the ZFNet network, the recognition rate progressively increases from 86.14% to 92.40% as the maximum iteration number augments. In contrast, for the VGG16 network, the difference in recognition rates between the 240,000 and 360,000 maximum-iteration models is marginal, with respective rates of 93.92% and 94.03%. Furthermore, the fluctuation of the loss function tends to plateau, indicating minimal improvement beyond this point. Consequently, to improve the computational efficiency, we limit the maximum number of iterations to 360,000. Ultimately, the vehicle brand detection and recognition network utilizes the VGG16 model trained for 360,000 maximum iterations.

The results pertaining to vehicle type recognition are presented in Table 2. Under the VGG16 network trained for 360,000 maximum iterations, the highest average recognition rate achieved is 97.62%. Notably, the recognition rates for buses and trucks attain exceptional levels of 98.86% and 99.56%, respectively, while the rates for other vehicle types are slightly lower. This can be attributed to the distinctiveness of bus and truck appearances, which facilitates easier classification. By balancing accuracy and computational efficiency, we opt to use the VGG16 network trained for 360,000 maximum iterations to effectuate vehicle type classification. This decision ensures both a high level of accuracy and a reasonable training time.

### 4.2. Comparison of Single-Angle and Multi-Angle Models

To ascertain the impact of multi-angle images on the model’s performance, this section undertakes a comparative analysis between models trained using single-angle and multi-angle images. Specifically, the VGG16 network was trained separately on each of the eight angles available in the Car5_48 dataset, namely: front (f), behind (b), right-front (rf), left-front (lf), left-side (ls), right-side (rs), right-behind (rb), and left-behind (lb). These models were then evaluated using multi-angle images. The resulting performance metrics are presented in Table 3. This empirical study allows us to quantify the influence of multi-angle images on the model’s ability to accurately detect and classify vehicle brands and to determine whether training on multi-angle images offers any advantages over training on single-angle images.

Table 3 presents a comparative analysis of the recognition rates achieved by the models trained on different angles. The results indicate that the models trained on specific angles, namely, rf, lf, rb, and lb, exhibit higher recognition rates compared to the other angles in the single-angle models. This can be attributed to the richer information content available in these angles. Notably, the model trained on images from all angles achieved the highest recognition rate of 95.33%, surpassing all other models that considered only a single angle. This observation underscores the importance of incorporating vehicle images from multiple angles during training, as it enables the network to capture more comprehensive information and enhance the accuracy of vehicle information detection models. Thus, leveraging multi-angle images can significantly improve the performance of deep learning models for vehicle detection tasks.

### 4.3. Vehicle Type Recognition Results

Experiments on vehicle type recognition were conducted across three datasets to classify five types of vehicles. The findings are presented in Table 4.

mAP1 denotes the average recognition rate across different vehicle types. As observed in Table 4, the truck achieved the highest accuracy (98.5%), closely followed by the bus (98.3%). The car and minivan achieved recognition rates of 95.0% and 93.3%, respectively, while the microbus and SUV had lower accuracy at 91.6% and 90.0%, respectively. The higher recognition rates for trucks and buses can be attributed to their distinct shape characteristics, which differentiate them from other vehicle types. Conversely, the lower accuracy for SUVs is primarily due to their similarity with other vehicle types, making them more challenging to distinguish.

mAP2 represents the vehicle type accuracy across different datasets. Analysis of Table 4 reveals that the Car5_48 dataset exhibited the highest recognition rate (97.62%), followed closely by the BIT-Vehicle dataset (94.1%). The Stanford Cars dataset had the lowest recognition rate at 88.3%. The superior performance of the car recognition network under the Car5_48 dataset can be primarily attributed to the similarity in the test and training sample collection environments and angles. This consistency allowed for a more accurate and robust classification of vehicle types.

Table 5 presents a comparative analysis of the recognition results on the BIT-Vehicle dataset. The proposed method achieved an improved recognition accuracy of 94.10%, which is 1.21% higher than the accuracy reported by Dong Z et al. [13] and 2.80% higher than that of Sang Jun et al. [15]. It is noteworthy that Dong Z et al. [13] utilized a convolutional neural network (CNN) for feature extraction, but their approach was limited to recognizing vehicle images from a single angle, thus discarding valuable detailed information. Similarly, Sang Jun et al. [15] employed a method similar to the one used in this study, but their detection was restricted to the front of the vehicle, which also resulted in a loss of more comprehensive details and compromised the robustness of the model to changes in vehicle angles. In contrast, the proposed method leverages multi-angle images to capture richer and more detailed feature information, enhancing the overall accuracy and robustness of the vehicle recognition system.

Figure 11 displays a representative selection of partial identification outcomes. The vehicle identification results depicted in the figure encompass a range of types, specifically including a bus, microbus, SUV, sedan, minivan, and truck. Notably, these results encompass images captured from diverse shooting angles, underscoring the robustness and adaptability of the identification system across various perspectives.

## 5. Conclusions

In this paper, a comprehensive multi-angle vehicle type and brand recognition method is constructed utilizing the Faster R-CNN framework. This innovative approach resolves the challenges associated with the multi-pose and multi-angle distribution of vehicle information recognition. Furthermore, to address the limitations of single-shot data collections in conventional datasets, a comprehensive vehicle type and brand dataset from eight diverse angles, designated as Car5_48, was created. Rigorous experimental evaluations demonstrate that the Faster R-CNN, when applied to multi-angle recognition, surpasses current state-of-the-art methodologies and enhances the overall robustness of the framework. This research contributes to the advancement of vehicle recognition techniques.

## Figures and Tables

**Figure 1 sensors-23-09569-f001:**
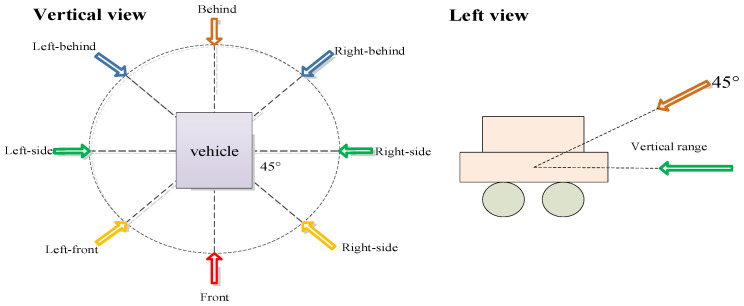
Schematic diagram of data collection angle.

**Figure 2 sensors-23-09569-f002:**
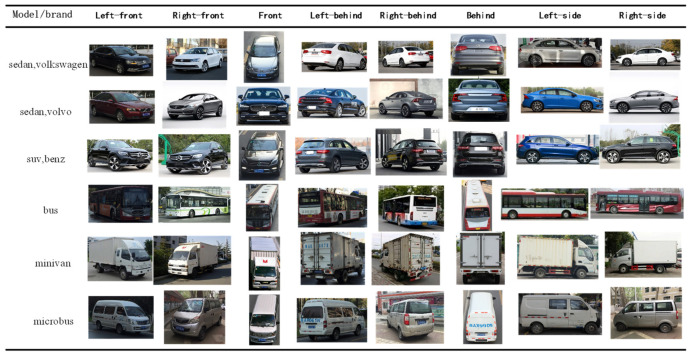
Example image of Car5_48.

**Figure 3 sensors-23-09569-f003:**
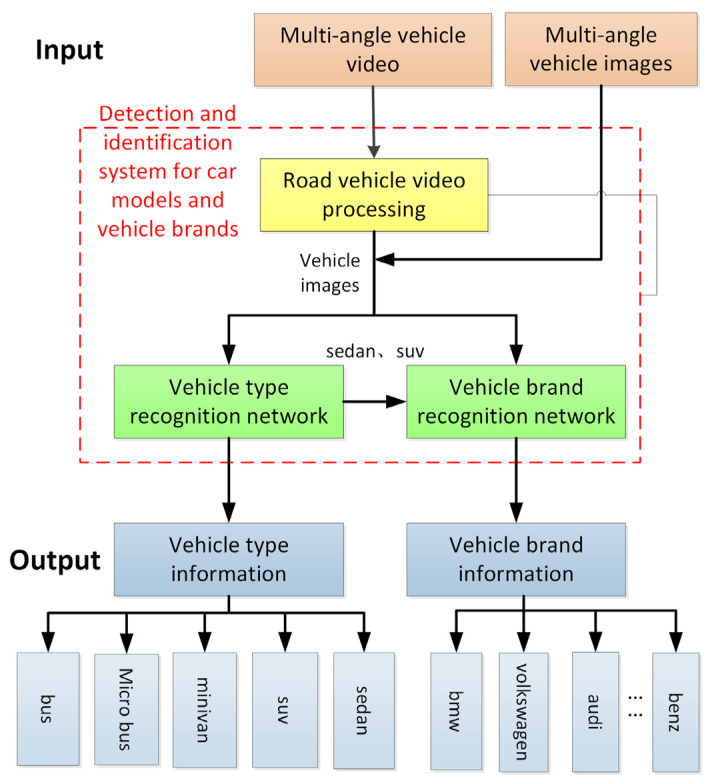
Overall framework.

**Figure 4 sensors-23-09569-f004:**
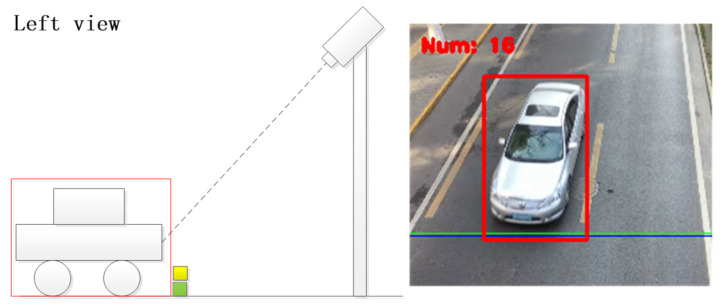
Image intercepted of a direct vehicle.

**Figure 5 sensors-23-09569-f005:**
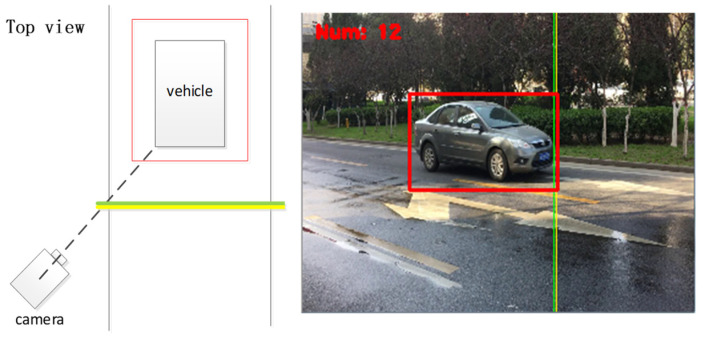
Image intercepted of a left-front vehicle.

**Figure 6 sensors-23-09569-f006:**
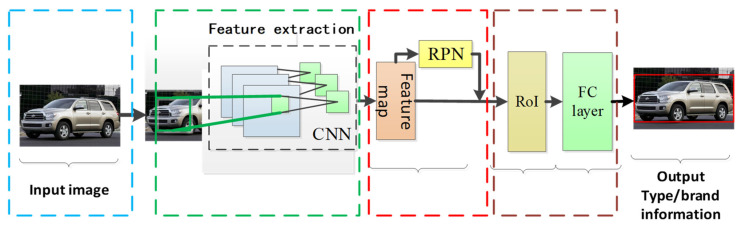
Architecture of the recognition network.

**Figure 7 sensors-23-09569-f007:**
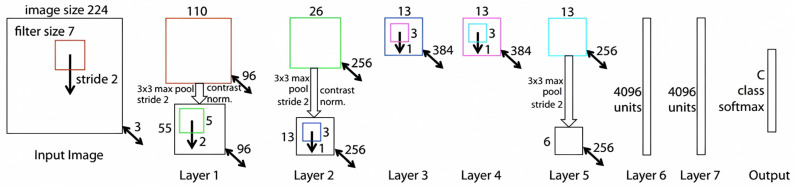
Architecture of ZFNet.

**Figure 8 sensors-23-09569-f008:**
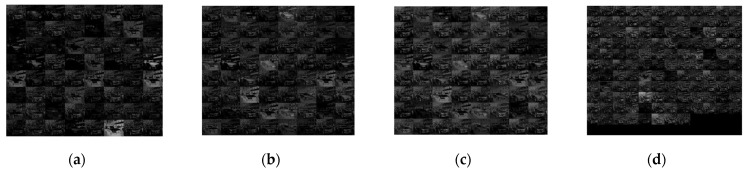
Examples of feature maps from convolutional layers (**a**,**c**) and pooling layers (**b**,**d**).

**Figure 9 sensors-23-09569-f009:**
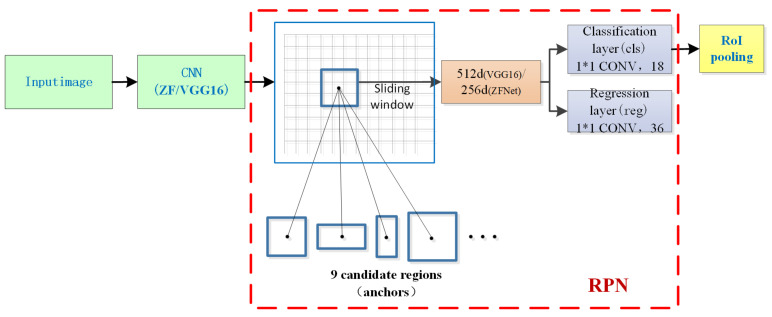
Architecture of the region proposal network.

**Figure 10 sensors-23-09569-f010:**
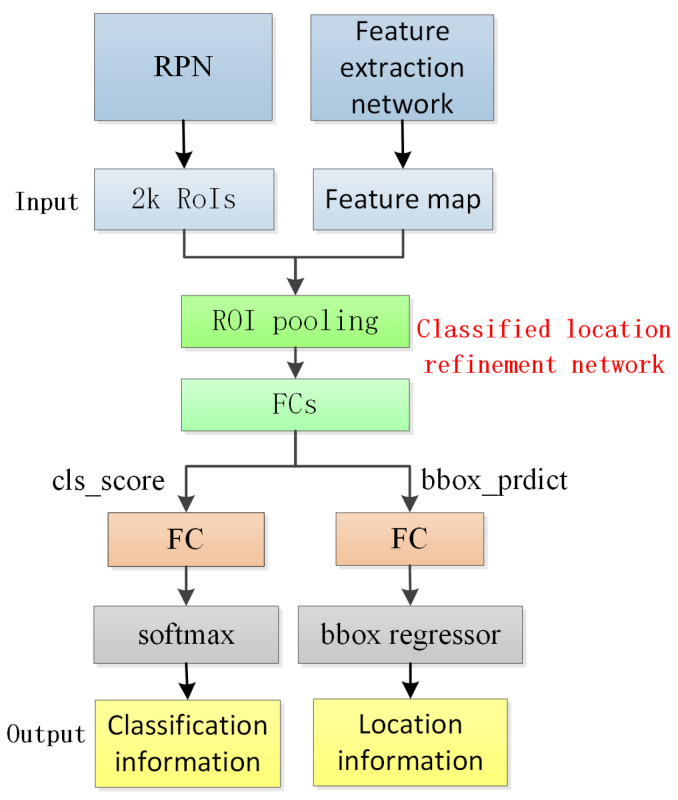
Classification location refinement network.

**Figure 11 sensors-23-09569-f011:**
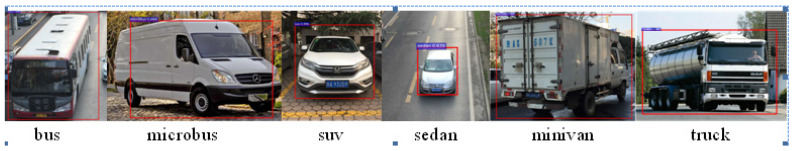
Example images of multi-angle vehicle brand position and recognition results.

**Table 1 sensors-23-09569-t001:** Vehicle brand recognition network results.

Feature Network	Maximum Iterations	Accuracy of Partial Vehicle					mAP
		BMW	Benz	BYD	Nissan	Land Rover	
ZFNet	240,000	0.9035	0.8495	0.8757	0.8788	0.8559	0.8614
ZFNet	360,000	0.9296	0.8611	0.8905	0.8798	0.9037	0.9240
VGG16	240,000	0.9561	0.8935	0.8950	0.8736	0.9434	0.9392
VGG16	360,000	0.9552	0.9174	0.9075	0.8806	0.9470	0.9403

**Table 2 sensors-23-09569-t002:** Vehicle type recognition network results.

Feature Network	Maximum Iterations	Accuracy						mAP
		Sedan	SUV	Bus	Minivan	Microbus	Truck	
ZFNet	240,000	0.9335	0.9128	0.9775	0.9258	0.9567	0.9810	0.9478
ZFNet	360,000	0.9407	0.9286	0.9856	0.9279	0.9537	0.9857	0.9612
VGG16	240,000	0.9725	0.9432	0.9834	0.9707	0.9689	0.9846	0.9705
VGG16	360,000	0.9764	0.9459	0.9886	0.9772	0.9735	0.9956	0.9762

**Table 3 sensors-23-09569-t003:** Comparison of experimental results from different angles.

Model	f	b	rf	lf	ls	rs	rb	lb	All Angles
Vehicle brand	0.8874	0.8650	**0.8995**	0.8863	0.8724	0.8955	0.8795	0.8720	0.9303
Vehicle type	0.9209	0.8922	0.9267	**0.9345**	0.8905	0.8562	0.9295	0.9059	0.9762
mAP	0.9042	0.8786	**0.9131**	0.9104	0.8815	0.8759	0.9045	0.8890	0.9533

**Table 4 sensors-23-09569-t004:** Multi-angle vehicle type recognition network test results.

Vehicle Model	Car5_48	Stanford Cars Dataset	BIT-Vehicle Dataset	mAP1
Sedan	0.9764	0.9000	0.9500	0.9500
SUV	0.9459	0.8500	0.9500	0.9000
Bus	0.9886	-	0.9670	0.9830
Minivan	0.9772	-	0.9330	0.9330
Microbus	0.9735	0.9000	0.9000	0.9160
Truck	0.9956	-	0.9750	0.9850
mAP2	0.9762	0.8830	0.9410	0.9340

**Table 5 sensors-23-09569-t005:** Comparison of recognition accuracy of different methods.

Methods	Recognition Angle	mAP
Dong Z [13]	Single angle	92.89%
Sang Jun [15]	Single angle	91.30%
The method in this part	Multi-angle	94.10%

## Data Availability

In this paper, three primary datasets are utilized: the Stanford Cars Dataset, the BIT-Vehicle Dataset, and the Car5_48 dataset. The Stanford Cars Dataset and the BIT-Vehicle Dataset are publicly available datasets, originating from academic publications and designed for research purposes. The Car5_48 dataset, on the other hand, is a self-built dataset specifically constructed for this study. This comprehensive dataset encompasses five distinct vehicle types and forty-eight different vehicle brands, providing a diverse and rich dataset for experimental analysis and research.

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
