# Peer review of "A Multi-Angle Appearance-Based Approach for Vehicle Type and Brand Recognition Utilizing Faster Regional Convolution Neural Networks"

_sensors, 2023, doi:10.3390/s23239569_

Round 1

Reviewer 1 Report

Comments and Suggestions for Authors

For the authors.

In Figure 1, characters appear most likely in Chinese; it is recommended that they be in English. Figure 2 shows “senda” instead of “sedan” in the model/brand column. Section 2.1 discusses the composition characteristics of data from another database, but needs to be more specific about the dataset features, since the 6 types and 48 proposed marks are not listed. In the same sense, Figure 3 shows 5 of the 6 classes; they should place all of them. Although placing the 48 marks in the figure is complicated, it would be ideal to mention them in some section of the article.

In section 3.1, particularly [126], based on other articles related to the topic of Gaussian distributions, we can infer that xt should be a pixel point of the image and in this section it is missing to specify that it belongs to a specific position. In [128], it is mentioned that K is assigned the value 4; however, in this type of articles it is recommended to explain the reason for this decision. The same goes for the value assigned to σi in [132] and 2.5 in equation (1), which again appears to be arbitrary. In addition, in [128] they place P(p), and perhaps it should be P(x).

Considering that [21] is the only reference that appears in this section (3.1), but we were not able to find the cited article [21] a greater rigor in the definition of variables is recommended, in addition to considering the use of more solid references.

In [238], “softmax classifier” is mentioned; however, it is important to note that softmax is a widely used activation function in classifiers, but it is not a classifier per se so it is incorrect.

In [269], iteration numbers are reported; however, it is not clear whether they are iterations across the large dataset or the number of iterated samples. It is recommended to report this data as the number of epochs that were used for such training.

From [314] to [319], it is necessary to specify whether it is a single algorithm trained with the various databases or if for each test a different model was chosen to train, since it is incorrect to indicate the percentage of accuracy for a dataset; The precision is obtained by the model, not the dataset.

Finally, in the conclusions section, the only model mentioned is Faster R-CNN; however, throughout the article VGG and ZFNetnare mentioned, but in the conclusions there is no comparative explanation of VGG 16, ZFNet and Faster R-CNN. On the other hand, during the article a detailed explanation of Faster R-CNN was not made. I believe that a better explanation of the use and comparison of results between VGG 16, ZFNet and Faster R-CNN is needed; otherwise, it does not justify the broad explanation of VGG 16 and ZFNet.

Comments on the Quality of English Language

Third, the article has a poor writing style that makes it difficult to understand. The author makes numerous spelling, grammatical and punctuation errors. I consider that the article requires revision by a native speaker of the English language.

Author Response

Reviewer 1:

Comments: In Figure 1, characters appear most likely in Chinese; it is recommended that they be in English. Figure 2 shows “senda” instead of “sedan” in the model/brand column. Section 2.1 discusses the composition characteristics of data from another database, but needs to be more specific about the dataset features, since the 6 types and 48 proposed marks are not listed. In the same sense, Figure 3 shows 5 of the 6 classes; they should place all of them. Although placing the 48 marks in the figure is complicated, it would be ideal to mention them in some section of the article.

Response:

(1). The characters in Fig. 1 have been modified;

(2). “senda” has been revised to “sedan”;

(3). The self-built dataset is actually Car5_48, which has been used to replace Car6_48.

 The following content has been added to the manuscript: "Specifically, these 5 types are cars, SUVs, buses, minivans, and minibuses, and the 48 brands are Volkswagen, BYD, BMW, Mercedes-Benz, Land Rover, Nissan, and so on."

(4). The listing of all 48 band names and the corresponding results are a bit cumbersome, and the recognition results of other brands are also similar. Therefore, we think the current discussion is sufficient.

Comments: In section 3.1, particularly [126], based on other articles related to the topic of Gaussian distributions, we can infer that xt should be a pixel point of the image and in this section it is missing to specify that it belongs to a specific position. In [128], it is mentioned that K is assigned the value 4; however, in this type of articles it is recommended to explain the reason for this decision. The same goes for the value assigned to σi in [132] and 2.5 in equation (1), which again appears to be arbitrary. In addition, in [128] they place P(p), and perhaps it should be P(x).

Response:

(1). The following content has been added to the manuscript: "In practical applications, we usually verify the effectiveness of initialization methods through experiments. Specifically, we can train multiple models using different initialization methods and compare their training and prediction performance. If a certain initialization method results in a model that performs well in both training and prediction, then we can consider this initialization method to be effective." From the final result, the initialization is shown to be effective.

(2) P(p) has been changed to be P(x).

Comments: Considering that [21] is the only reference that appears in this section (3.1), but we were not able to find the cited article [21] a greater rigor in the definition of variables is recommended, in addition to considering the use of more solid references.

Response:

(1). Section 3.1 has been rewritten, and all the variables are described.

Comments: In [238], “softmax classifier” is mentioned; however, it is important to note that softmax is a widely used activation function in classifiers, but it is not a classifier per se so it is incorrect.

Response:

(1). “softmax classifier” has been replaced by "“softmax".

(2). The following content has been added to the manuscript: "The Softmax function is a commonly used function in deep learning, especially when dealing with multi-class classification problems. It maps a set of real values to a probability distribution, where each element of the output result is between 0 and 1, and the sum of all elements is equal to 1."

Comments: In [269], iteration numbers are reported; however, it is not clear whether they are iterations across the large dataset or the number of iterated samples. It is recommended to report this data as the number of epochs that were used for such training.

Response:

(1). “iteration" has been changed to "epoch".

Comments: From [314] to [319], it is necessary to specify whether it is a single algorithm trained with the various databases or if for each test a different model was chosen to train, since it is incorrect to indicate the percentage of accuracy for a dataset; The precision is obtained by the model, not the dataset.

Response:

(1). “iteration" has been changed to "epoch"

(2) The backbone models ZFNet and VGG are tested for our proposed dataset.

Comments: Finally, in the conclusions section, the only model mentioned is Faster R-CNN; however, throughout the article VGG and ZFNet are mentioned, but in the conclusions there is no comparative explanation of VGG 16, ZFNet and Faster R-CNN. On the other hand, during the article a detailed explanation of Faster R-CNN was not made. I believe that a better explanation of the use and comparison of results between VGG 16, ZFNet and Faster R-CNN is needed; otherwise, it does not justify the broad explanation of VGG 16 and ZFNet.

Response:

(1). Faster R-CNN is composed of a back-bone network and other parts, and both VGG and ZFNet are the back-bone nets. From Table 1 and 2, we conclude that the VGG back-bone is better, therefore the final architecture of Faster R-CNN is composed of VGG and other parts.

Comments: Third, the article has a poor writing style that makes it difficult to understand. The author makes numerous spelling, grammatical and punctuation errors. I consider that the article requires revision by a native speaker of the English language.

Response:

(1). The whole paper has been rewritten, even for the title and the abstract.

Reviewer 2 Report

Comments and Suggestions for Authors

Aiming at the problem of ignoring the multi-attitude and multi-angle distribution of vehicles, this paper identifies vehicle models. I think the innovation is not high enough and the significance is not great. Please elaborate on the significance and purpose of this research and what kind of scenarios it can be applied to.

Comments on the Quality of English Language

1. There are some grammatical mistakes and poor sentence structure in the article.

2. Some sentences are too long and need to be broken down or reorganized.

3. Some less common abbreviations are used in the article and may need to be explained or replaced with more common words.

4. Some sentences are not clear enough and need to be further clarified or reorganized.

Author Response

Reviewer 2:

Comments: Aiming at the problem of ignoring the multi-attitude and multi-angle distribution of vehicles, this paper identifies vehicle models. I think the innovation is not high enough and the significance is not great. Please elaborate on the significance and purpose of this research and what kind of scenarios it can be applied to.

Response:

(1). Despite considerable efforts expended on this problem, the effectiveness of existing solutions remains limited. A key challenge in vehicle information recognition stems from the wide array of vehicle models and designs across different brands, coupled with the rapid variation in appearance as the viewing angle changes over time. This complexity demands a more sophisticated and robust approach to accurately discern and classify vehicle information in diverse real-world scenarios.

Comments:

1.There are some grammatical mistakes and poor sentence structure in the article.

  1. Some sentences are too long and need to be broken down or reorganized.
  2. Some less common abbreviations are used in the article and may need to be explained or replaced with more common words.
  3. Some sentences are not clear enough and need to be further clarified or reorganized.

Response:

(1). The whole paper has been rewritten, even for the title and the abstract.

Reviewer 3 Report

Comments and Suggestions for Authors

The article proposes a method for vehicle type and brand recognition based on Faster R-CNN, which can handle multi-angle images of vehicles. It also introduces a novel dataset called Car6_48, which contains six types and 48 brands of vehicles from eight angles under different environments. Before the paper could be published, I believe the following questions need to be addressed:

1.      The paper should provide a detailed description of the Car6_48 dataset, including how it was collected and annotated, its advantages and disadvantages, and why it is suitable for evaluating the proposed method. Is there any plan for Publishing the Car6_48 dataset?

2.      The proposed pipeline requires multiple input images than a single image pipeline. Can you comment on the pros and cons?

3.      Also, from raw camera captured images, which might include many cars, don’t you need some computation module to pre-process the data so that the inputs of the network will be all from the same car?

4.      How does the video processing model handle occlusion, noise, and illumination changes in the video frames? How does it affect the performance of the recognition network?

5.      How does the choice of feature extraction network (ZFNet or VGG16) affect the results of vehicle type and brand recognition? What are the trade-offs between them?

6.      Minor: Can you change the Chinese characters in Figure 1. to English?

Author Response

Reviewer 3:

Comments:

  1. The paper should provide a detailed description of the Car6_48 dataset, including how it was collected and annotated, its advantages and disadvantages, and why it is suitable for evaluating the proposed method. Is there any plan for Publishing the Car6_48 dataset?

Response:

(1). The following content has been added to the manuscript: "Using the CCD camera of Yuanda Youshi (YDV-C932A) to capture and collect training and testing vehicle driving video samples. When capturing vehicle driving video, a camera is placed in front of and on the side of the road, and the vehicle is driven straight ahead to obtain images from four angles: straight ahead, left, left front, and left rear. When the vehicle turns around, images are obtained from four angles: straight behind, right, right front, and right rear. In addition, the shooting environment and weather conditions are as diverse as possible, in order to obtain vehicle samples under different lighting conditions. For manually captured vehicle samples, in order to diversify the background conditions, a large number of shooting locations are selected."

(2) We are delighted to release this dataset later.

Comments: 2.      The proposed pipeline requires multiple input images than a single image pipeline. Can you comment on the pros and cons?

Response:

(1). By processing multiple images simultaneously, the model can leverage contextual information between images, thereby improving the accuracy of object detection.

(2). When the target object is occluded in one image, information from other images can be used to assist detection, which helps to solve occlusion problems.

(3). Processing multiple images requires more computational resources, including memory and computing time, which may lead to reduced training and inference speeds.

(4). Multi-image input pipelines require more complex data processing and model design, potentially increasing the difficulty of development and debugging.

Comments:3.      Also, from raw camera captured images, which might include many cars, don’t you need some computation module to pre-process the data so that the inputs of the network will be all from the same car?

Response:

(1). We don't think it's a good idea to use different images of the same car as input. This will reduce the generalization ability of the model.

Comments: 4.      How does the video processing model handle occlusion, noise, and illumination changes in the video frames? How does it affect the performance of the recognition network?

Response:

(1). Occlusion: Occlusion occurs when an object or a part of it is blocked by another object in the scene. To handle occlusion, the video processing model may use techniques such as: using spatial and temporal information to predict the location and movement of occluded objects; employing multiple cameras or viewpoints to overcome occlusions by providing additional information from different angles; utilizing contextual information from the surrounding environment to infer the presence and location of occluded objects.

(2). Noise: Noise in video frames can be caused by various factors such as camera sensor imperfections, compression artifacts, or transmission errors. To handle noise, the video processing model may use techniques such as: image denoising algorithms, and temporal filtering by combining information from adjacent frames.

(3). Illumination changes: Variations in lighting conditions can significantly affect the appearance of objects in video frames, making it challenging for recognition networks. To handle illumination changes, the video processing model may use techniques such as: histogram equalization to adjusts the contrast of an image, making it more uniform and improving the visibility of details; normalizing the brightness and color of video frames, to help make the recognition network more robust to illumination changes; extract features that are less sensitive to illumination changes, making the recognition network more robust.

(4). The performance of the recognition network is affected by how well the video processing model handles these challenges. If the model is successful in mitigating occlusion, noise, and illumination changes, the recognition network is likely to perform better in terms of accuracy and reliability. Conversely, if these issues are not adequately addressed, the recognition network's performance may suffer, leading to lower accuracy, increased false positives or false negatives, and reduced overall system performance.

Comments: 5.    How does the choice of feature extraction network (ZFNet or VGG16) affect the results of vehicle type and brand recognition? What are the trade-offs between them?

Response:

(1). In the Faster R-CNN network, we use the same hyperparameters, but only use ZFNet and VGG in the backbone part, and use the self-built dataset Car5_48 to evaluate the detection performance. The results show that VGG as a feature extraction network can achieve better performance. Therefore, in the subsequent discussion of this article, VGG is selected to form the final Faster R-CNN network.

Comments: 6.  Minor: Can you change the Chinese characters in Figure 1. to English?

Response: Figure 1 has been modified.

Round 2

Reviewer 1 Report

Comments and Suggestions for Authors

Thank you for addressing the comments from the previous review.

There are still slight aspects that I believe can be changed. I list them below:

[82, 469] six types are mentioned, however, you have already made the change to use only five types. 
[176] a space is required between 'n' and 'is'.
[193] may be miss-placed or there is a lack of context.
[200] use 'bounding box' instead of 'bounding rectangle'
[281, 343] 'SoftMax loss' is incorrect as We have already discussed before this is an activation function.
[330] please consider using 'improve' instead of 'optimize'.
[359] please validate if the number of epochs reported is correct, taking into account that an epoch is an iteration over the entire dataset, and the number reported is very large. 
[434, 437] please add the number of the reference (if is already cited).

Comments on the Quality of English Language

There are words in English that are either stilted or not appropriate to the context, please consider replacing them. A few examples are listed below:

[31] realm
[95] avenues
[224] undergo
[375] forego
[463] paves

Author Response

Thank you very much for taking the time to review this manuscript.Please refer to the attachment below for detailed response content.

Reviewer 3 Report

Comments and Suggestions for Authors

I appreciate the authors' efforts in addressing the majority of the concerns I raised; however, I would like to re-emphasize the importance of Comment 3. This comment was not meant to offer a training method for the network but rather to highlight a potential obstacle encountered during the inference phase. The images used in your dataset depict idealized scenarios, typically featuring a single car per image. This representation does not reflect the complexity of real-world conditions where images often contain multiple vehicles within the same frame. I am skeptical about the network’s ability to process such multifaceted real-world images effectively, given that it has been trained on these 'ideal' cases. Could you elaborate on how you might adapt your preprocessing to accommodate these more complex scenarios? Additionally, what would be the anticipated computational cost associated with implementing such a preprocessing module?

Author Response

Thank you very much for taking the time to review this manuscript. Please refer to the attachment below for detailed response content.

Round 3

Reviewer 3 Report

Comments and Suggestions for Authors

The authors have addressed the comment. I recommend the paper for publication.